# Single-cell sequencing reveals lineage-specific dynamic genetic regulation of gene expression during human cardiomyocyte differentiation

Reem Elorbany[1]☯, Joshua M. Popp[2]☯, Katherine Rhodes[3], Benjamin J. Strober[2], Kenneth Barr[3], Guanghao Qi[2], Yoav Gilad[3,4]*, Alexis Battle[2,5,6]*

1 Interdisciplinary Scientist Training Program, University of Chicago, Chicago, Illinois, United States of America, 2 Department of Biomedical Engineering, Johns Hopkins University, Baltimore, Maryland, United States of America, 3 Department of Human Genetics, University of Chicago, Chicago, Illinois, United States of America, 4 Department of Medicine, University of Chicago, Chicago, Illinois, United States of America, 5 Department of Computer Science, Johns Hopkins University, Baltimore, Maryland, United States of America, 6 Department of Genetic Medicine, Johns Hopkins University, Baltimore, Maryland, United States of America

☯ These authors contributed equally to this work.
* gilad@uchicago.edu (YG); ajbattle@jhu.edu (AB)

**Data Availability Statement:** All expression data files, pre- and post-processing, are available from the GEO database (accession number

## Abstract

Dynamic and temporally specific gene regulatory changes may underlie unexplained genetic associations with complex disease. During a dynamic process such as cellular differentiation, the overall cell type composition of a tissue (or an *in vitro* culture) and the gene regulatory profile of each cell can both experience significant changes over time. To identify these dynamic effects in high resolution, we collected single-cell RNA-sequencing data over a differentiation time course from induced pluripotent stem cells to cardiomyocytes, sampled at 7 unique time points in 19 human cell lines. We employed a flexible approach to map dynamic eQTLs whose effects vary significantly over the course of bifurcating differentiation trajectories, including many whose effects are specific to one of these two lineages. Our study design allowed us to distinguish true dynamic eQTLs affecting a specific cell lineage from expression changes driven by potentially non-genetic differences between cell lines such as cell composition. Additionally, we used the cell type profiles learned from single-cell data to deconvolve and re-analyze data from matched bulk RNA-seq samples. Using this approach, we were able to identify a large number of novel dynamic eQTLs in single cell data while also attributing dynamic effects in bulk to a particular lineage. Overall, we found that using single cell data to uncover dynamic eQTLs can provide new insight into the gene regulatory changes that occur among heterogeneous cell types during cardiomyocyte differentiation.

GSE175634). Genetic data for the Yoruba cell lines used are available at zenodo.org/record/5826177#.YdhKCRPMJb8. The code for this analysis is available on Github at github.com/jmp448/sc-dynamic-eqtl/.

**Funding:** Y.G. and A.B. were supported by NIH/NIGMS R01GM120167 (https://www.nigms.nih.gov/Research/mechanisms/Pages/researchprojectgrants.aspx). A.B. was supported by NIGMS R35GM139580 (https://www.nigms.nih.gov/research/mechanisms/mira/pages/default.aspx). R.E. was supported by the NIH MSTP Training Grant T32GM007281 (https://www.nigms.nih.gov/training/instpredoc/pages/predocoverview-mstp.aspx). J.P. was supported by NIH/NIGMS T32GM119998 (https://www.icm.jhu.edu/academics/graduate-curricum/pre-doctoral-training-program-in-computational-medicine/). K.R. was supported by NIH/NHLBI 5F31HL146171 (https://researchtraining.nih.gov/programs/fellowships/f31). The funders had no role in study design, data collection and analysis, decision to publish, or preparation of the manuscript.

## Author summary

Many complex traits and diseases are associated with genetic variants which are suspected to regulate the expression levels of nearby genes. However, we are still unable to identify many of the relevant variant-gene associations. Previous work has shown that regulation of gene expression is often specific to a biological context, suggesting that measuring gene expression in diverse contexts may reveal important associations. In this work, we identified genetic regulatory effects that are "dynamic" over time in a complex environment containing diverse and transient cell states. We collected single-cell gene expression data at several time points from cells differentiating, or changing state, from stem cells to cardiomyocytes. We characterized two distinct trajectories that cells undertake as they differentiate *in vitro*, and assigned each cell to a particular point along a specific trajectory. We then identified hundreds of dynamic associations between regulatory variants and gene expression levels, including many specific to a single trajectory. This work demonstrates the importance of searching for variant-gene associations in cell types that change over time or exist only during fleeting stages of cellular differentiation, and provides a framework for identifying these associations in the presence of bifurcating trajectories that are characteristic of human development.

## Introduction

A primary aim of human genetics and genomics is to understand the genetic architecture of complex traits. Current studies demonstrate that the majority of trait-associated genomic loci are in non-coding regions of the genome, and are thought to be involved in gene regulation [1]. Therefore, studies exploring gene regulation are essential to our understanding of complex phenotypes [2,3]. Studies mapping expression quantitative trait loci (**eQTLs**), identifying genetic variants associated with gene expression levels, reveal the impact of genetic variation on gene regulation and can inform molecular mechanisms underlying trait-associated loci. eQTLs have now been identified for a wide variety of tissues, and their study has contributed to the understanding of gene regulation and disease [4–10].

Gene regulation, including genetic regulation of gene expression, can vary between contexts including different cell types, temporal stages, and environmental stressors. Particular attention has been paid to differences in gene regulation between tissues and cell types. Large studies including the Genotype-Tissue Expression Project (**GTEx**) have been now been successful in identifying thousands of eQTLs in diverse human tissues [4,11]. However, despite these efforts, we are still unable to identify a regulatory mechanism for the genetic contribution of a majority of disease-associated loci [12–16]. One reason for this knowledge gap may be that most large-scale eQTL studies are based on expression data from adult, bulk tissue samples that do not represent the specific cell types and contexts in which disease-relevant dysregulation occurs [17].

Recent advances in single-cell sequencing have allowed us to assay gene expression in individual cells, allowing us to access disease relevant cell types and cell states, even if they compose a small fraction of a tissue and would not be well captured by bulk data, and even if they are not known a priori. Indeed, single cell datasets have revealed a more complex landscape of gene expression in individual cell types than previously known in tissues such as brain and kidney [18,19]. Likewise, mapping eQTLs from single-cell RNA-sequencing data promises to enable the identification of previously undiscovered disease-relevant regulatory mechanisms.

Recently, collection and analysis of population-scale scRNA-seq datasets have demonstrated that genetic effects do vary between cell types belonging to the same tissue [20–22].

Beyond cell-type specificity, only a small number of studies have attempted to characterize dynamic gene regulatory changes that occur during development or among contexts that change over time [23–30]. These have highlighted temporally specific eQTL effects that were not evident from static data. Studying the temporal dynamics of gene expression has the potential to uncover genomic loci involved in gene regulation during developmental processes and identify associations that were previously overlooked. Accordingly, we previously studied genetic effects on the regulation of gene expression during the differentiation of induced pluripotent stem cells (**iPSCs**) to cardiomyocytes [23]. We collected time-series bulk RNA-seq data for nineteen individuals to identify hundreds of eQTLs displaying dynamic, and sometimes transient effects on expression across the course of cardiomyocyte differentiation. These dynamic eQTLs included genetic variants which were associated with cardiovascular disease-related traits, including obesity.

However, the complexities of cardiomyocyte differentiation and other dynamic processes are not fully captured by bulk RNA-seq data even in a time course study design. During development and differentiation, expression profiles change over time in individual cells along a spectrum of maturity [31]. Cells within a single sample do not necessarily differentiate at the same rate, along the same trajectory, or even toward the same terminal cell type. Different cell lines may also vary in the proportion of cells in different states at each time point. Indeed, in our previous work, we identified two clusters of cell lines undergoing cardiomyocyte differentiation that exhibited broad differences in the expression trajectory of groups of genes over time [23]. Bulk expression profiles represent an average across cells from various points across a developmental landscape, obscuring the underlying variation in cell state, and even making it difficult to definitively attribute differences to cis-regulatory genetic effects. Recent work has demonstrated that the improved resolution of single-cell RNA-seq data can identify homogeneous subpopulations of cells at similar stages of differentiation, offering a clearer view of genetic regulation in an individual time step [32,33]. However, such analysis has only been applied to a few cell types, not including cardiomyocytes, and has been limited to the study of dynamics within a single lineage.

In this study, we applied single-cell RNA-seq to the nineteen cell lines assayed in our previous bulk RNA-seq analysis, collecting single-cell data at seven informative time points during cardiomyocyte differentiation, enabling us to observe cell-type specificity, cell composition differences, and temporal changes together in a unified experiment. The resolution of this single cell data enables us to characterize the cardiomyocyte differentiation landscape in much greater detail than was possible in bulk. We identify a bifurcation in cell fate, which explains the previously observed clustering of cell lines and enables us to study genetic regulatory dynamics along two distinct trajectories with a single experiment. Characterization of these trajectories allows us to reanalyze existing bulk samples and mitigate confounding impact of cellular composition and identify dynamic effects specific to each lineage [22,34].

## Results

We differentiated induced pluripotent stem cells (**iPSCs**) from 19 human cell lines into cardiomyocytes; these same cell lines were previously used for a cardiomyocyte time course study published in Strober et al 2019 [23]. For the current study, we used new iPSC cultures of the same lines, and differentiated them again to cardiomyocytes. We used Drop-seq to collect single-cell RNA-seq data at 7 days throughout the 16-day differentiation time course. We chose to collect data from days 0 (iPSC), 1, 3, 5, 7, 11, and 15 (cardiomyocyte), as we have previously

observed that these days represent the most informative stages during this particular differentiation trajectory [23,35]. We collected single-cell data using a balanced study design in which each collection included three individuals at three unique differentiation time points. While cells from the same cell line and differentiation day have similar expression profiles, likely due to biological factors, this study design minimizes technical effects associated with collection batch (**S1 Table** and **S1 Fig**, Materials and Methods). After filtering data from low quality cells (Materials and Methods), the resulting 131 samples contained an average of 1,762 cells per sample and an average of 1,375 genes detected as expressed per cell. Following normalization, a principal component analysis revealed that, as expected, differentiation day is the primary axis of variation in the single cell gene expression data (**S2 Fig**).

## Differentiation progress and cell line differences drive variation in gene expression

In order to characterize the complex landscape of cardiomyocyte differentiation, we used UMAP to produce a low-dimensional embedding of the single cell data while preserving global structure. We found that while cells from the early days of the differentiation time course exhibited fairly uniform transcription profiles, this was less true for later days (days 7, 11, and 15; **Fig 1A** and **1D**). Marker genes known to be expressed at various stages in cardiac differentiation, from iPSC to mesoderm to cardiomyocyte, showed high expression at expected early, intermediate, and late stages of the differentiation time course, respectively (**Fig 1B** and **1E**). Next, we used unsupervised clustering to partition cells into clusters, and matched cell clusters to known cell types based on expression of known marker genes (**Fig 1C**, Materials and Methods, [36]). While these discrete cell type assignments are imperfect as they fail to capture the continuity of the differentiation process, they are useful in characterizing the broad relationships between groups of cells. As suggested by previous reports [23,35], we identified a bifurcation in the differentiation landscape, giving rise to two distinct terminal cell types. One of these terminal cell types has high expression of genes known to be involved in cardiomyocyte function, such as *TNNT2* and *MYL7* (**Fig 1B**, [37,38]). Cells in the other terminal cell type do not express cardiomyocyte markers, and instead have high expression of genes such as *COL3A1* and *VIM*, which are expressed in the extracellular matrix of cardiac fibroblasts [39,40]. The differentiation outcome of each sample, namely the proportion of cells in each cluster, varied by individual cell line; certain lines differentiated primarily into either the *TNNT2*-expressing or the *COL3A1*-expressing terminal cell type clusters (**Figs 1E** and **S3**). For the remainder of this paper, we will refer to the *TNNT2*-expressing cell cluster as cardiomyocytes (CM) and to the *COL3A1*-expressing cluster as cardiac-fibroblasts (CF) or fibroblast. We also identified a cluster that underexpressed marker genes of cardiac cell types throughout the differentiation process, and instead expressed several endoderm-specific markers such as *APOA1* and *AFP*. We were unable to fully characterize this cluster based on expression patterns alone, and omitted these cells from downstream investigation of the dynamics of gene regulation on gene expression during mesoderm and cardiac cellular differentiation (**S4 Fig**).

## Single-cell expression data resolves bifurcating trajectories during cellular differentiation

In previous work, we investigated the relationship between genotype and chronological time, represented by the differentiation day in which each bulk sample was collected. However, chronological time may not properly capture the axis of variation along which genetic regulation is changing, and can be heavily confounded by heterogeneity in differentiation within and between samples. If cells within a sample progress through differentiation at different rates,

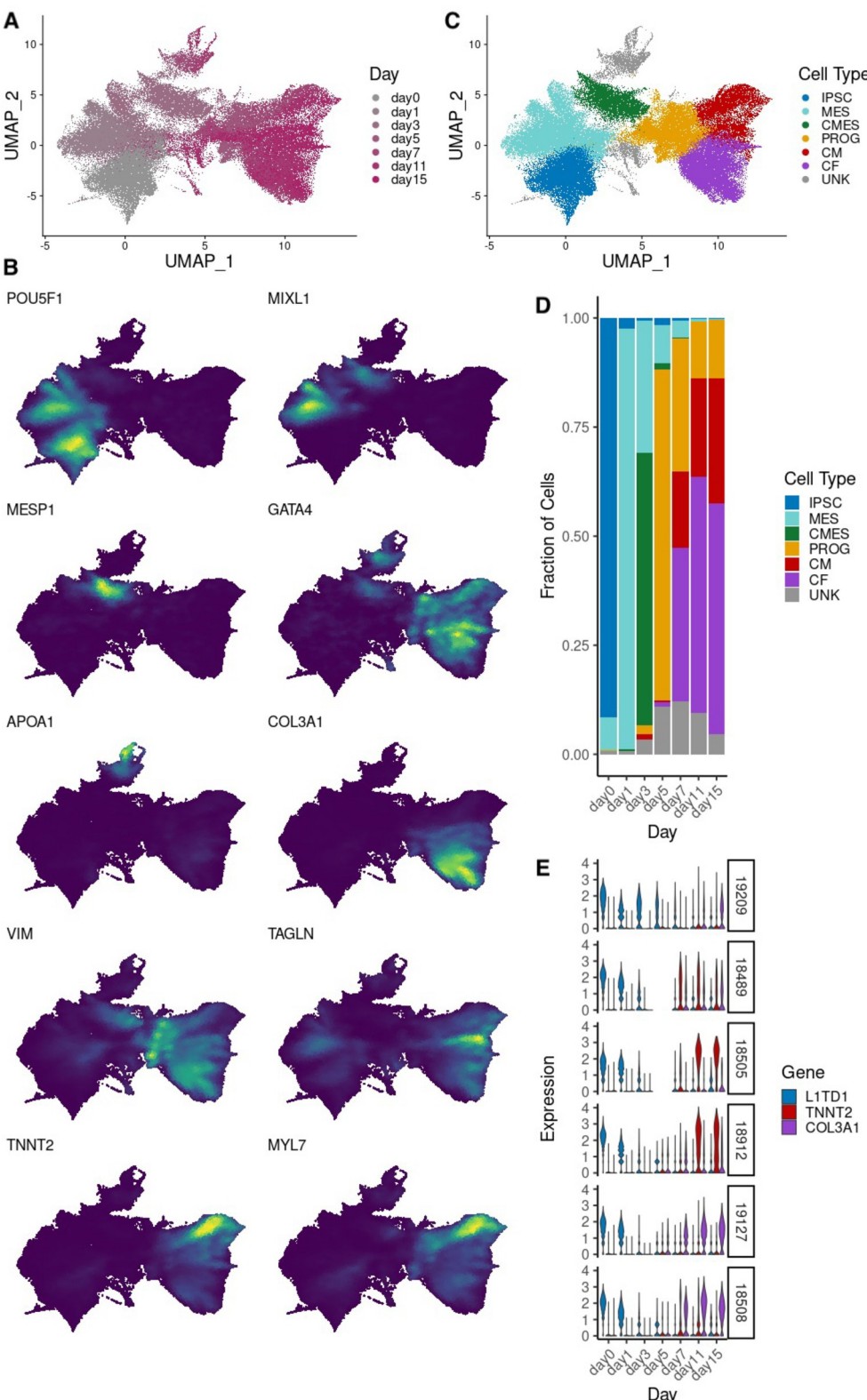

**Fig 1. Gene expression patterns in single cell data.** (A) UMAP of full single cell dataset; cells are colored by differentiation day. (B) Estimated density of expression for several marker genes across cells. (C) UMAP of full single cell dataset; cells are colored by cell type, assigned based on Leiden clustering and marker gene expression.

IPSC = induced pluripotent stem cell, MES = mesoderm, CMES = cardiac mesoderm, PROG = cardiac progenitor, CM = cardiomyocyte, CF = cardiac fibroblast, UNK = unknown cell type. (D) Proportion of cells belonging to each cell type per differentiation day, across all cell lines. (E) Distribution of *L1TD1* (pluripotency marker), *TNNT2* (cardiomyocyte marker) and *COL3A1* (cardiac fibroblast marker) over cells from 6 representative examples of the 19 cell lines studied, for each of the 7 differentiation days.

their aggregated expression profile will not be truly reflective of an individual stage of differentiation, confounding tests for association between genotype and differentiation progress. Systematic differences between cell lines can exaggerate this: differentiation speed appears to vary between cell lines (Fig 1E), such that differentiation progress at day 3, for example, is not uniform across samples. Such differences can lead to false associations between genotype and differentiation progress in cases where genotype is partially correlated with a cell line's differentiation speed.

Cellular heterogeneity drives further challenges when aggregating across cells that are differentiating along diverging paths. Aggregated bulk profiles will lose information about the individual cell types present, and if cell type composition varies between individuals (Fig 1E), this will further confound associations between genotype and expression changes during differentiation.

By collecting expression at the single-cell level, we are able to address both of these challenges. To properly focus on the two primary cardiac lineages present, we used the *scanpy* package to produce a low-dimensional Force Atlas embedding of the cells that had been successfully assigned to a known cell type (**Fig 2A**, Materials and Methods, [41,42]). We inferred pseudotime for each cell with diffusion pseudotime [43,44], so that progress through differentiation is learned from cells' individual expression profiles rather than their time of collection (**Fig 2B**). We performed trajectory analysis using PAGA [44] to examine the relationships between the cell types present in this dataset, which helped to resolve two distinct lineages present in the data, giving rise to cardiomyocyte and cardiac fibroblast cell types (**S5 Fig**).

One disadvantage to single-cell data compared to bulk is that single-cell measurements are more sparse and noisy: by aggregating over cells, bulk RNA-sequencing reduces noise, which makes expression measurements more tractable for eQTL calling. This introduces a tradeoff between the flexibility of analysis at the single cell level, where we can explore a broader range of dynamic effects among finely resolved pseudotimes or cell populations, and the robustness of analysis on aggregated data [45]. To balance this tradeoff, we partitioned cells from each lineage into pseudotime bins, pooling information across cells to mitigate the noisiness of single cell expression measurement while maintaining homogeneous populations of cells through lineage subsetting and pseudotime binning. Cells were assigned based on the trajectory analysis to the cardiomyocyte lineage, the cardiac fibroblast lineage, or both in the case of precursor cell types. This aggregation scheme enables us to produce a greater number of samples, as we are no longer constrained to the 7 days when experimental collection was performed, while maintaining the expected trends of lineage-specific marker gene expression over pseudotime (**Fig 2C-2F**).

## Mapping of dynamic eQTLs

We applied a Gaussian linear model to the aggregated single-cell pseudo-bulk data based on pseudotime bins from each lineage to identify dynamic eQTLs, namely variant-gene pairs in which the interaction effect of genotype and differentiation time is significantly associated with changes in gene expression. We identified linear dynamic eQTLs for 357 genes in the cardiomyocyte lineage ($q < 0.05$) and 903 genes in the cardiac fibroblast lineage (Materials and Methods; **Table 1**). The difference in the number of dynamic eQTLs detected between the two

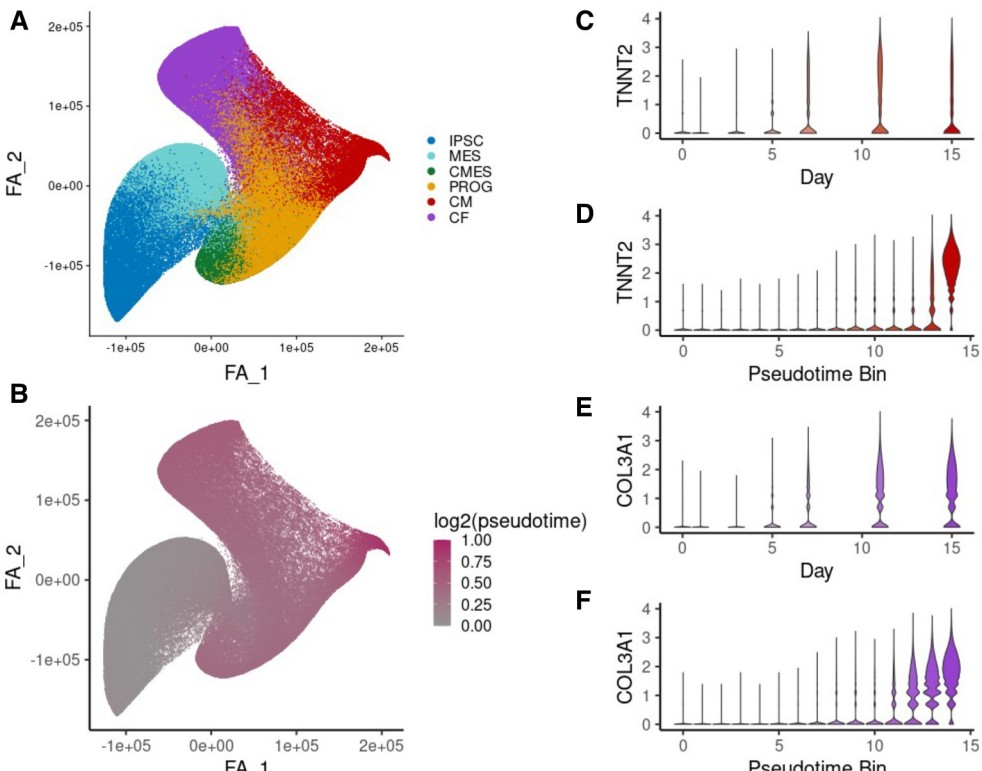

**Fig 2. Pseudotime inference and pseudobulk aggregation.** (A) Force atlas embedding of all cells from the two cardiac differentiation lineages combined, colored by cell type. IPSC = induced pluripotent stem cell, MES = mesoderm, CMES = cardiac mesoderm, PROG = cardiac progenitor, CM = cardiomyocyte, CF = cardiac fibroblast, UNK = unknown cell type. (B) Force atlas embedding from (A), colored by $log_2$(pseudotime+1), which was inferred for each cell shown using diffusion pseudotime. (C) Distribution of normalized expression of *TNNT2*, a cardiomyocyte marker gene, across cells from the cardiomyocyte lineage for each differentiation day. (D) Normalized *TNNT2* expression across cells from each of 16 pseudotime quantile bins along the cardiomyocyte trajectory. (E) Normalized expression of *COL3A1*, a cardiac fibroblast marker, across cells from the cardiac fibroblast lineage for each differentiation day. (F) *COL3A1* expression across cells for 16 pseudotime quantile bins along the cardiac fibroblast trajectory.

lineages may arise due to greater heterogeneity among predominantly cardiomyocyte samples, or from a difference in the number of cells captured (45,980 cardiac fibroblasts versus 21,862 cardiomyocytes) leading to more precise pseudobulk expression profiles in predominantly

**Table 1. Comparison of linear dynamic eQTL calling methods.** We report the number of linear dynamic eGenes (genes with a significant dynamic eQTL at gene-level q-value < = 0.05), for each of the aggregation schemes assessed. Total number of genes tested and total number of tests run are also reported.

| Dataset | Aggregation | Time Points | Lineage | Linear Dynamic eGenes Detected | Total # Genes Tested | Total # Tests |
|---|---|---|---|---|---|---|
| Pseudobulk | Pseudotime | 16 | CM | 357 | 8,969 | 1,601,727 |
| Pseudobulk | Pseudotime | 16 | CF | 903 | 9,140 | 1,633,408 |
| Pseudobulk | Differentiation Day | 7 | CM | 142 | 9,541 | 1,693,532 |
| Pseudobulk | Differentiation Day | 7 | CF | 100 | 9,548 | 1,711,693 |
| Pseudobulk | Differentiation Day | 7 | Combined | 5 | 9,656 | 1,731,798 |
| Bulk | Differentiation Day | 7 | Combined | 210 | 10,772 | 1,963,378 |
| Bulk | Differentiation Day | 16 | Combined | 1028 | 10,981 | 1,991,072 |

cardiac fibroblast samples. We found that both lineage specificity and the replacement of real chronological time with pseudotime improved power for dynamic eQTL detection: using chronological differentiation day as the time variable identified only 142 and 29 dynamic eQTLs for the cardiomyocyte and cardiac fibroblast lineages, respectively. Using differentiation day as the time variable and omitting lineage specificity altogether identified only 5 dynamic eQTLs in the pseudobulk data. For comparison, an analysis of static (non-dynamic) eQTLs using the *mashr* framework [46], which similarly aggregated pseudobulk by cell line and differentiation day, revealed 183 static eQTLs (Materials and Methods).

Ultimately, our lineage subsetting and pseudotime approach revealed slightly more dynamic eQTLs than were previously identified in an experiment with bulk collections at over twice as many time points [23]. To ensure a meaningful comparison, we re-processed the previously collected bulk data in a similar pipeline as pseudo-bulk, accounting for changes in hypothesis testing and filtering of variant-gene pairs (Materials and Methods). This revealed a total of 1028 genes with a dynamic eQTL (compared to a total of 1056 genes detected between both lineages with pseudobulk binned to a similar number of samples; S6 Fig). When the bulk data was subset to the same 7 collection time points used for the single cell experiment, only 210 dynamic eGenes were detected. The increased detection rate offered by the pseudobulk analysis may stem from increased homogeneity of cellular populations that undergo pseudo-bulk aggregation, as well as improved measurement of differentiation progress achieved by using cellular pseudotime rather than sample collection time.

We investigated several potential sources of confounding in this analysis. First, we assessed calibration of dynamic eQTL calling by permuting the pseudotime variable in the interaction term before calling dynamic eQTLs, which did not reveal evidence of substantial inflation (Materials and Methods, S7 Fig). Second, we investigated the possibility that broad differences between cell lines, such as variation in differentiation speed or trajectory preference, are driving false positive discoveries. We did not find evidence of confounding among dynamic eQTLs, as would be suggested by elevated pairwise correlation of genotype at these loci (Materials and Methods, S8 Fig). This is likely because regression of cell line principal components (Materials and Methods, [23]) effectively controls for such broad differences (S9 Fig). Third, we checked for type I error inflation due to 'double dipping', where the hypotheses tested are influenced by analysis of the data itself [47–49], since pseudotime is inferred from single-cell expression data. We demonstrate in simulation that the fixed-effect linear model used in this study was conservative in the presence of multiple measurements per individual and did not lead to type I error inflation, though there may be some loss in sensitivity (Materials and Methods, S10 Fig). Finally, we investigated the impact of uncertainty in pseudotime estimation by examining the impact of permuting cellular pseudotime before pseudobulk aggregation, which did not suggest inflation (Materials and Methods, S11 Fig).

As an example of the trait relevance of these dynamic eQTLs, one dynamic eQTL variant, rs1234988, has previously been implicated by GWAS to be associated with hypertension (p = 2.5e-35), and was detected as a dynamic eQTL for *ARHGAP42*, a Rho GTPase which has previously been identified as a critical regulator of vascular tone and hypertension in mice (**Figs 3A, 3B** and **S12**, [50,51]). Notably, *ARHGAP42* is known to be a smooth-muscle selective Rho GAP, and this dynamic eQTL was exclusively identified in the cardiac fibroblast lineage (Bonferroni-adj. p = 2.4e-5, cardiac fibroblast lineage, adj. p = 0.79, cardiomyocyte lineage). This variant is not detected as a dynamic eQTL without lineage subsetting or pseudotime binning (adj. p = 1). This example illustrates the advantages of incorporating exploratory data analysis in the study of *in vitro* experimental datasets: while the differentiation procedure used for these experiments was designed to produce exclusively cardiomyocytes, an alternative

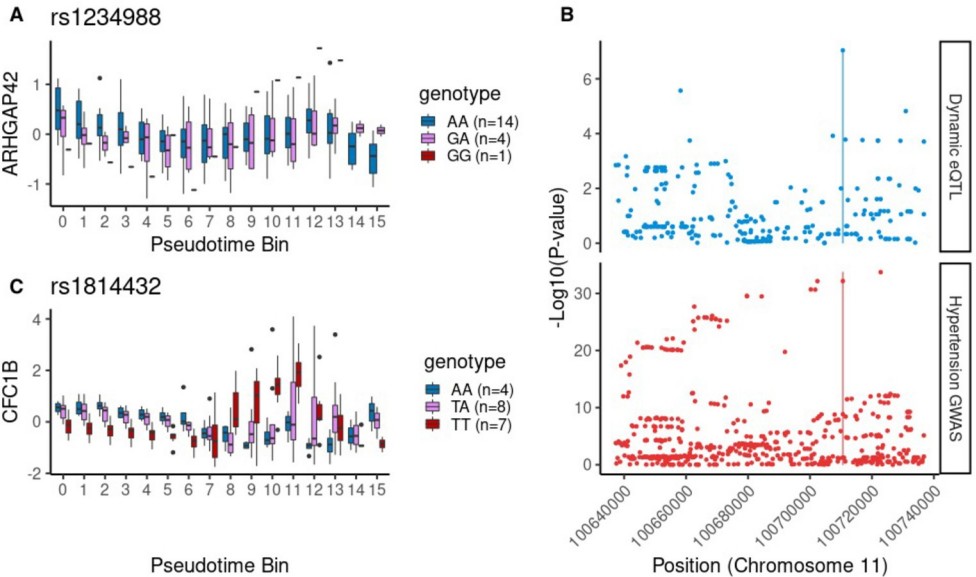

**Fig 3. Linear and nonlinear dynamic eQTLs.** (A) rs1234988 is a linear dynamic eQTL for *ARHGAP42*; the effect of genotype (color) on *ARHGAP42* expression (y-axis) varies across pseudotime (x-axis). (B) A previously reported genome-wide association study (bottom) showed that hypertension is associated with genotype at the rs1234988 locus, where a dynamic eQTL for *ARHGAP42* was identified. (C) rs1814432 is a nonlinear dynamic eQTL for the gene *CFC1B*.

terminal cell type discovered after exploratory data analysis is able to provide meaningful insight into an additional differentiation process.

The pseudotime values can be interpreted as intermediate time points with greater resolution than chronological time. We therefore used these values to also identify nonlinear dynamic eQTLs, whose effects vary in a nonlinear way over the course of differentiation, such as presence only at intermediate stages of the differentiation (**Figs 3C** and S12). We identified 74 nonlinear dynamic eQTL variants for the cardiomyocyte lineage (q<0.05), and 147 for the cardiac fibroblast lineage (Materials and Methods; **S2 Table**). Our time course study design is particularly useful for detecting transient nonlinear genetic effects which may not be found by studying only the initial or terminal cell types of a dynamic process such as differentiation.

We examined the extent to which the dynamic eQTLs detected overlapped with eQTLs previously identified in GTEx [4]. After subsetting to gene-variant pairs that were tested in both our data and GTEx, we found that the greatest replication of pseudotime-binned, cardiomyocyte lineage linear dynamic eQTLs occurred in atrial appendage tissue ($\pi_1 = 0.50$, method described in [52]), while the greatest replication of pseudotime-binned, cardiac fibroblast linear dynamic eQTLs (as well as bulk) occurred in cultured fibroblasts ($\pi_1 = 0.47, 0.56$ respectively). However, by searching directly for dynamic effects across cell types rather than within a single tissue in isolation, we additionally identify eQTLs which were not found to be a significant eQTL in any tissue in GTEx. After subsetting to variant-gene pairs that were tested in both our data and GTEx, we found that 100 of the 359 (28%) linear dynamic eQTLs in the cardiomyocyte lineage were identified as eQTLs in GTEx. Similarly, only 22 of 75 (29.3%) nonlinear dynamic eQTLs on the cardiomyocyte lineage where previously identified as eQTLs in GTEx. Further classification of these cardiomyocyte lineage nonlinear dynamic eQTLs as "early-acting", "late-acting", "transient" (or "middle"), and "switch" (as in ref. [23]) found that most of these variants (68 of 74) are late-acting, supporting the similar replication rates between linear and nonlinear dynamic eQTLs. Of the five early-acting and transient dynamic eQTLs, none were identified as cis-eQTLs in GTEx.

## Deconvolution of bulk RNA sequencing data assigns lineage specificity to dynamic eQTLs

The information about the landscape of cardiomyocyte information obtained through single-cell RNA sequencing can also be applied to improve dynamic eQTL calling in bulk data. For each cell type that we identified in the single cell data, we computed a signature expression profile across the top 300 differentially expressed genes that were also measured in bulk (Materials and Methods). We then used CIBERSORTx to deconvolve our bulk data, assigning to each bulk sample a vector of cell type proportions (**Figs 4A** and **S13**, Materials and Methods, [53]). Deconvolution reveals that cell type heterogeneity is prominent between samples,

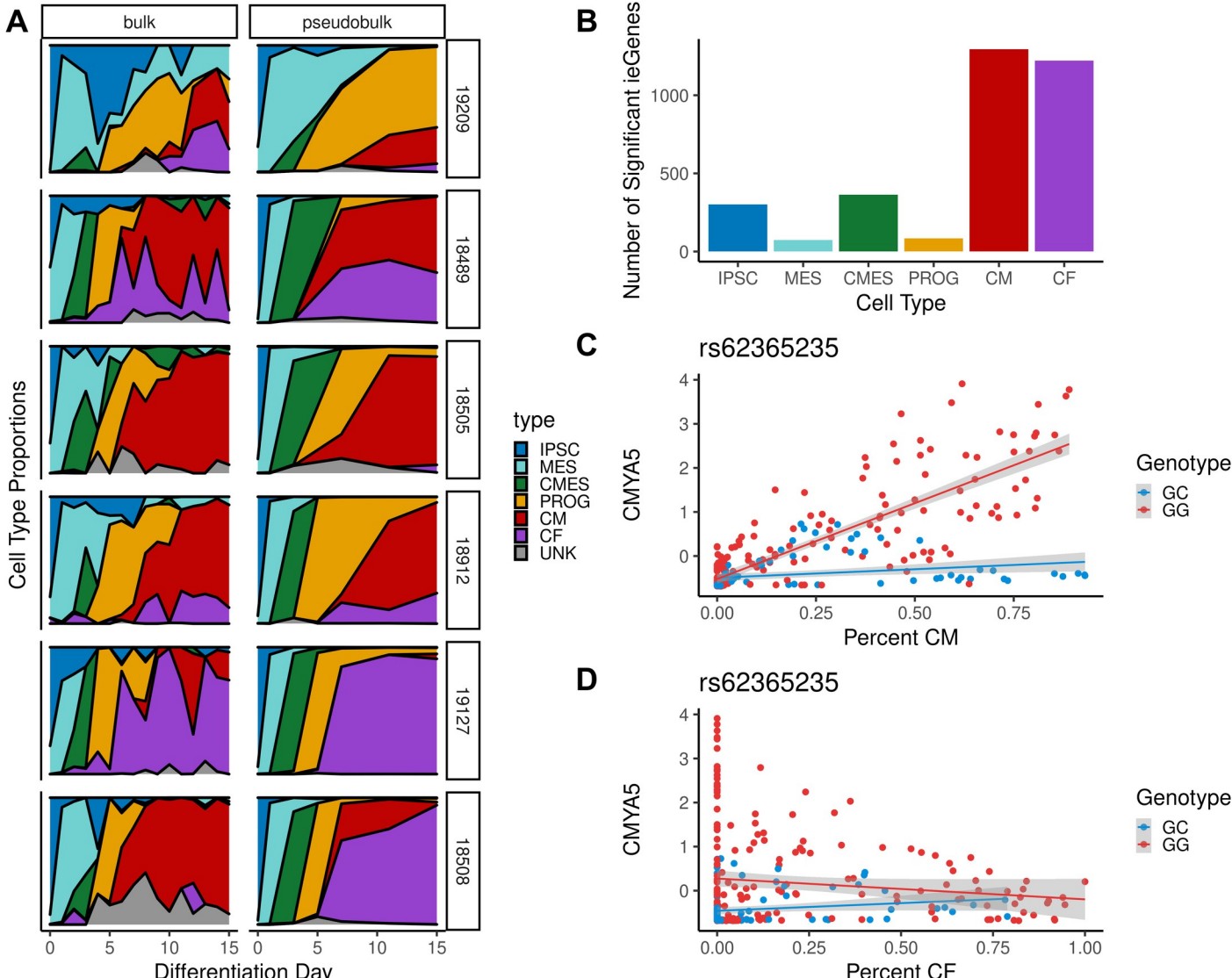

**Fig 4. Cell type deconvolution and interaction eQTL calling.** (A) Cell type deconvolution was applied to decompose RNA expression of a mixed sample, aggregated over multiple cell types, into its constituent cell type proportions (Materials and Methods). Each row represents a cell line, collected in two separate experiments. In the left column, bulk RNA-sequencing data was collected for 15 timepoints (time on x-axis). In the right column, pseudobulk was aggregated across cells collected for 7 time points (time on x-axis). For pseudobulk data, deconvolution is not needed, as each cell is assigned to a cell type. Thus, "ground truth" cell type fractions are accessible as reflected here. (B) Number of genes with a cell type interaction eQTL in bulk for each of six cell types. (C-D) *CMYA5* has an interaction eQTL for the cardiomyocyte lineage (C) that is not identified in the cardiac fibroblast lineage (D).

particularly in days 7–15. This heterogeneity emphasizes the need to account for cell type proportion in measuring genetic regulatory dynamics, as these broad differences between cell lines can drive false positive associations between time and any genotype that is correlated with broad cell type proportion differences between cell lines.

We then used these cell type proportions to identify cell type specific regulatory effects, based on cell type interaction eQTLs (**ieQTLs**) for each known cell type that was observed in the single cell data (**Fig 4B**). Interaction eQTLs for a cell type at an endpoint of the differentiation (iPSC, cardiomyocyte [CM], and cardiac fibroblast [CF]) are analogous to linear dynamic eQTLs, since cellular composition often partially reflects differentiation time. However, this relationship varies between lineages: we find that cardiomyocyte proportion is more correlated with differentiation day than cardiac fibroblast proportion (Pearson's $\rho = 0.59$ for cardiomyocyte proportion, compared to $\rho = 0.36$ for cardiac fibroblast proportion). Accordingly, we found that CM and CF ieQTLs called with this approach were replicated in the previously used dynamic eQTL calling framework on the same bulk dataset to varying degrees ($\pi_1 = 0.84$ and 0.43, respectively). This may reflect a fundamental difference between the two differentiation trajectories, if cells within samples which produce primarily cardiac fibroblasts reach maturity more quickly. This hypothesis is supported by the observation that the maximum pseudotime value of the cardiomyocyte lineage is greater than that for the cardiac fibroblast lineage (Fig 2B).

This approach enables insights that could not be attained using bulk data alone. The use of an expression signature matrix derived from single cell data with multiple terminal cell types allows us to characterize hundreds of lineage-specific effects which were previously obscured in bulk data: we find that many of the CM- and CF-ieQTLs are lineage-specific (78% and 77% respectively), including some which are potentially relevant to heart-related disease. **Fig 4C and 4D** show an example of a cardiomyocyte interaction eQTL for cardiomyopathy-associated protein 5 (*CMYA5*), a gene which is highly expressed in heart and skeletal muscle and has previously been associated with cardiac hypertrophy [54]. This variant was not previously identified by GTEx as an eQTL for *CMYA5*. More broadly, these interaction eQTLs showed enrichment for genes related to myogenesis that had not been observed among bulk dynamic eQTLs (p = 7e-4, both CM and CF ieQTL, compared to p = 0.17, bulk dynamic eQTL).

This paradigm for analysis is potentially powerful, for example, given bulk data with larger sample sizes or denser time point sampling possible due to lower cost and effort of bulk sequencing, combined with smaller-scale single cell data that offers cell type, lineage, and pseudotime resolution. Given the denser time-point sampling in our bulk data, there are many interaction QTLs that were not detected in the single-cell data alone, or bulk data alone, but enabled by the integration of bulk and single-cell data together.

## Discussion

Using iPSCs and their derived terminal cell types, we can identify genetic effects related to dynamic changes in gene expression over time. We used single-cell gene expression data to investigate the effects of gene regulatory and cell type composition changes throughout a cardiomyocyte differentiation time course. Single-cell data enables us to identify cells going down distinct differentiation trajectories, and to deconvolve heterogeneous cell types in matched bulk samples.

One question that arises from these single-cell data is the interpretation of distinct differentiation trajectories and potentially different cell types at the end of the time course. We found that, in later stages of differentiation (days 7, 11, and 15), most cells have either high gene expression of cardiac troponin T (*TNNT2*) and associated genes such as myosin light chain/

*MYL7*, or high gene expression of a collagen-coding gene (*COL3A1*) and associated genes such as vimentin/*VIM*, as discovered through a semi-supervised pipeline which includes dimensionality reduction, unsupervised clustering, and visualization of expression patterns for known marker genes (Fig 1B). Cells broadly express either of these gene sets in a mutually exclusive manner, suggesting that these gene sets represent two distinct cell types. The focus of this project was not to fully characterize these cell types, but instead to disentangle the broad effects of cell line differences in differentiation rate/lineage preference from the dynamics of cis-regulation of gene expression. Still, the identity of these terminal cell types and the circumstances in which each trajectory might be favored is an interesting question.

These data suggest that there are differences in gene expression trajectory and ultimate cell fate that may arise in response to the same differentiation protocol. The identity of these terminal cell types, and the factors that might cause a cell line to favor one differentiation trajectory and ultimate cell type at the expense of another, are questions that have been explored in previous studies. In a study by D'Antonio-Chronowska et al. [55], embryonic stem cell lines undergoing cardiac differentiation resulted in a heterogeneous cell type population. These cells were identified as either true cardiomyocytes—which exhibit mechanical beating and have high expression of *TNNT2*—or "epicardium-derived cells" which do not exhibit mechanical beating and have high expression of gene markers such as *VIM* and *TAGLN*. The study demonstrated that these two cell types were present in varying proportions in each individual cell line, and suggests that this cell fate decision can be influenced by genetic factors, such as variability in X chromosome gene dosage [55].

The cardiomyocyte and epicardium framework explored by D'Antonio-Chronowska et al. [55] may be useful in understanding the distinct differentiation trajectories present in our cardiac differentiations. The terminal non-cardiomyocyte cells expressing *COL3A1* in these samples may represent an endothelial or cardiac fibroblast cell type, which derive from the epicardium cell lineage. Cardiac fibroblasts express gene markers such as collagen and vimentin, which were found to be expressed in the terminal cells of this differentiation trajectory [39,40,56]. The gene expression profile of *COL3A1*-expressing cells, which includes high expression of genes related to extracellular matrix and physical cellular structure, implies that these terminal cells may be involved in providing some kind of structural support, perhaps as a reinforcement to true beating cardiomyocytes.

To determine whether differentiation trajectory and ultimate cell fate decision is influenced by genetic factors, it may be useful to perform cardiomyocyte differentiation with multiple replicates of each cell line, and compare the differentiation trajectories between these replicates. The relatively high correlation between these single-cell RNA-seq samples compared to matched bulk RNA-seq samples of the same cell line [23] suggests that there may be genetic factors involved in this trajectory decision—although more rigorous testing should be performed to investigate this claim. We may also investigate whether subtle systematic differences exist between cell lines even in the iPSC stage (Day 0), and whether these differences correlate with the ultimate trajectory of these cell lines during differentiation. Recent studies have suggested that there may be genes whose expression level at the iPSC stage correlates with downstream differentiation efficiency in a predictable manner [32,33]. Their results suggest that the decision for ultimate cell type trajectories remains consistent within a cell line, and that iPSCs from those cell lines exhibit distinct gene expression profiles that can be used to accurately predict differentiation trajectories even before differentiation begins. This is an intriguing possibility, and more work should be performed to investigate whether the cell lines used here also exhibit distinct gene expression profiles early on that may correlate with the outcomes of any subsequent differentiation.

While simulations suggested that inferring pseudotime before testing for dynamic genetic effects did not result in type I error inflation in this case (S10 Fig), the potential pitfalls of this type of 'double dipping' will remain an important consideration for future analyses and applications. Existing methods which address this type of circularity [49] rely on resampling procedures that pose a bottleneck when combined with rapidly advancing and often computationally expensive unsupervised or semi-supervised machine learning tools, including many popular pseudotime inference tools.

All together, the results from this study demonstrate the benefit of using single-cell RNA-sequencing with a balanced time course study design to investigate dynamic gene regulatory differences between individuals during cellular differentiation. Single-cell data offers a high-resolution view of the landscape of differentiation, which we leveraged to infer pseudotime along multiple differentiation trajectories. By isolating axes of variation of cis-regulatory dynamics (pseudotime within a particular lineage, rather than chronological differentiation day), we were able to identify a greater number of dynamic eQTLs with less than half as many collection time points as previous efforts in bulk RNA-seq data. The dynamic eQTLs detected included variants which overlapped known GWAS hits, demonstrating the utility of this approach in identifying causal loci that underlie risk for development of disease. We also used this data to lend new utility to bulk RNA-seq datasets, by assigning lineage specificity to dynamic eQTLs through the use of cell type interaction eQTL calling. While further follow-up studies should be performed to validate the function of these genomic loci and their potential relevance to downstream phenotypes, the dynamic genetic effects identified in this study and the methodology used to identify them provide a resource for investigating mechanisms underlying important biological processes such as cellular differentiation and perturbation response.

## Materials and methods

### Ethics statement

The cell lines used in this study were obtained from the NHGRI Sample Repository for Human Genetic Research at the Coriell Institute for Medical Research. All samples were collected by the Coriell Institute for Medical Research with written informed consent and with IRB approval. The genetic data used here has previously been made available through the International HapMap Project [57].

### Samples

We used induced pluripotent stem cell (iPSC) lines from 19 individuals from the Yoruba Hap-Map population. These iPSC lines were reprogrammed from lymphoblastoid cell lines and characterized previously [58]. All 19 individuals were female and unrelated. We chose to use only female individuals to avoid introducing additional variance that is not of interest in this study.

### iPSC maintenance

Feeder-free iPSC cultures were maintained on Matrigel Growth Factor Reduced Matrix (CB40230, Thermo Fisher Scientific, Waltham, MA) with Essential 8 Medium (A1517001, Thermo Fisher Scientific) and Penicillin/Streptomycin (30002Cl, Corning, Corning, NY). Cells were grown in an incubator at 37˚C, 5% CO2, and atmospheric O2. Cells were passaged to a new dish every 3–5 days using a dissociation reagent (0.5 mM EDTA, 300 mM NaCl in PBS) and seeded with ROCK inhibitor Y-27632 (ab120129, Abcam, Cambridge, UK).

## Cardiomyocyte differentiation

We differentiated iPSCs using a protocol previously optimized for use with the Yoruba Hap-Map panel [58]. This protocol implements slight modifications to the cardiomyocyte differentiation protocols from [59] and [36]. Feeder-free iPSCs were seeded onto wells of a 6-well plate and grown for 3–5 days prior to differentiation. When most lines were 70%-100% confluent, E8 media was replaced with "heart media" along with 1:100 Matrigel hESC-qualified Matrix (08-774-552, Corning) and 12uM of GSK-3 inhibitor CHIR99021 trihydrochloride (4953, Tocris, Bristol, UK). "Heart media" is composed of RPMI (15-040-CM, Thermo Fisher Scientific) with B27 Supplement minus insulin (A1895601, Thermo Fisher Scientific), 2mM Gluta-MAX (35050–061, Thermo Fisher Scientific), and 100mg/mL Penicillin/Streptomycin (30002Cl, Corning). CHIR99021 is a small molecule that activates WNT signaling and initiates the differentiation on day 0 (after the 'day 0' cell collection) [59]. "Heart media" was replaced 24 hours later at day 1 of differentiation. 48 hours later, at day 3 of differentiation, cells were fed with new "heart media" containing 2uM of the WNT inhibitor Wnt-C59 (5148, Tocris) [59]. We cultured cells in Wnt-C59 heart media for 48 hours. At day 5, Wnt-C59 was removed, and base "heart media" was added. "Heart media" was refreshed on days 7, 10, 12, and 14 of differentiation. Cells began spontaneous mechanical beating between days 7 and 13 of differentiation.

In some cases, after performing cardiac differentiation, one might choose to perform a post hoc purification process to remove any non-cardiac cell types present at the terminal time point [60]. However, for the purposes of a time course experiment where multiple intermediate time points are assayed, a purification protocol undertaken only at the end of the differentiation would not prove useful; therefore, no cell type purification was performed.

## Sample collection and processing

We performed cardiomyocyte differentiations in three total batches of six to seven cell lines at a time. For each batch, cardiomyocyte differentiations were performed with three staggered starting days, such that samples could be collected from each cell line in three differentiation stages at any given time. For all 19 cell lines, samples were collected on differentiation days 0 (iPSC, before treatment with CHIR99021), 1, 3, 5, 7, 11, and 15. Drop-seq collection was performed a total of three collection days for each batch of six to seven cell lines. In the first collection day, samples from all cell lines in the batch were collected for differentiation days 1, 3, and 7. In the second collection day, samples from all cell lines in the batch were collected for differentiation days 5, 7, and 11. In the third collection day, samples from all cell lines in the batch were collected for differentiation days 0 (iPSC), 11, and 15. Through this process, single-cell gene expression data was collected for all cell lines in seven unique time points, with two time points (differentiation days 7 and 11) having two replicates. This staggered differentiation and collection study design was performed to minimize the technical effect of sample collection as a potential confounding variable associated with cell line or differentiation day.

To harvest the samples at the start of each collection day, cells in at least two wells of a 6-well culture dish were released from the dish using Accutase (BD Biosciences, San Jose, CA, #561527). Samples were washed three times and resuspended in 1X PBS, 0.01% BSA. Cells were then passed through a 40 um filter to encourage the formation of a single cell suspension. The concentration of each single cell suspension was quantified manually using an NI hemocytometer (INCYTO, Cheonan, Korea, DHC-N01-2).

Using a 125 um Drop-seq microfluidic device, single cells were captured in droplets along with a DNA barcoded bead (ChemGenes, Wilmington, MA, Macosko-2011-10(V+)), following the standard Drop-seq protocol [61]. The DNA barcoded beads include a cell-specific

barcode so the cell identity of each RNA molecule can be recovered. After Drop-seq collection, the RNA molecules were reverse transcribed, and cDNA amplification was performed according to the Drop-seq protocol. cDNA concentration and library size were measured using the Qubit 3 fluorometer (Thermo Fisher) and BioAnalyzer High Sensitivity Chip (Agilent, Santa Clara, CA, #5067–4626).

Library preparation was performed using the Illumina Nextera XT DNA Library Preparation Kit (Illumina, FC-131-1096). Libraries in each batch were multiplexed together so that every sequencing lane contained three samples, one from each of the three collection days. Each of those samples was itself a multiplexed collection of three individual cell lines at three distinct differentiation time points, which were mixed upon Drop-seq collection. Samples went through paired-end sequencing using the Illumina NextSeq 500. 20 bp were sequenced for Read 1, and 60 bp for Read 2 using Custom Read 1 primer, GCCTGTCCGCGGAAGCAGTGGTATCAACGC AGAGTAC, according to manufacturer's instructions [61]. The same multiplexed library pool was sequenced twice with the goal of achieving at least 20 million reads per sample.

We recorded 20 technical and biological covariates and measured their contribution to variation in our data (**S14 Fig**).

## RNA-seq quantification

For each sequencing run, we obtained paired-end reads, with one pair representing the cell-specific barcode and unique molecular identifier (UMI), and the second pair representing a 60 bp mRNA fragment. We used dropseqRunner (available at github.com/aselewa/dropseqRunner) which takes a fastq file with paired-end reads as input and produces an expression matrix corresponding to the UMI of each gene in each cell. All RNA-seq samples were aligned to the human genome (GRCh38) using STAR-solo [62]. We used featureCounts [63] to assign each aligned read to a genomic feature, and umi_tools [64] to create a count matrix representing the frequency of each feature in our dataset. We then used the single-cell demultiplexing software 'demuxlet' to assign to each cell a probability that the cell is a doublet [65].

The following filter was applied to remove 21,725 rare genes (out of 60,668) from downstream analysis:

- Gene must be detected in at least 10 cells

The following filters were then applied to remove 330,750 low-quality cells (out of 564,362) for downstream analysis:

- Maximum doublet probability of 0.3 from demuxlet

- Unambiguous assignment of the cell to an individual by demuxlet (maintain cells not assigned to 'doublet_ambiguous')

- Maximum of 25% mitochondrial reads

- Minimum of 300 unique genes detected (of the genes that passed the previous filtering step)

Following these filtering steps, an additional 2,826 cells were removed whose feature or read counts were more than 4 standard deviations away from the median. This left a total of 230,786 cells and 38,943 genes for downstream analysis (**S15–S17 Figs**).

## Cell cycle correction and normalization of single-cell expression data with Seurat

We used the Seurat workflow for cell cycle regression in differentiating. Each cell was assigned a score for G2/M phase and S phase according to marker gene expression, and the difference

between these scores was regressed out during normalization. The data was then normalized using the SCTransform function in [66,67], producing corrected counts, log-normalized corrected counts, Pearson residuals, and a set of highly variable features. The Pearson residuals of 1,000 highly variable features were scaled so that each gene had unit variance across all cells for downstream analysis.

### Dimensionality reduction and clustering with scanpy

Dimensionality reduction, clustering and pseudotime were performed using the *scanpy* package [41], following Seurat object to h5ad conversion via the *sceasy* package [68]. The scaled Pearson residuals from 1000 highly variable features were used to compute 50 principal components (PCs), which were then embedded into a 2D UMAP plot (Fig 1A and 1C). These 50 PCs were also used to produce a neighborhood graph, and Leiden clustering was performed at resolution 0.35 to produce the clusters shown in Fig 1C. (Several clusters are merged into the unknown cell type, as described below).

### Lineage specification and pseudotime inference

Based on marker gene expression patterns (Fig 1B), 6 of the 10 Leiden clusters were annotated with known cell types. To facilitate trajectory reconstruction, 3 outlier clusters with less than 5,000 cells were removed. Cluster 7 contained a group of cells which did not express marker genes for cardiomyocytes or progenitor cell types, and instead expressed a group of genes that are specifically expressed in hepatocytes, a cell type stemming from the endoderm layer rather than the mesoderm layer. This small population of cells drove a significant amount of variation in the data (S4 Fig), making it difficult to properly resolve the mesoderm-specific lineages that were the focus of this project. For this reason, the cells assigned to one of the mesoderm-specific lineages (clusters 1–6) were used for downstream analysis (a UMAP embedding of the subset data is shown in **S18 Fig**).

The log-normalized gene expression was re-centered and re-scaled, and PCA was re-run on specifically these cells to properly focus on the variation among the lineages of interest. The top 3 re-computed PCs were used to calculate a new neighborhood graph, which was used to compute a new embedding to visualize specifically the two cardiac-related differentiating lineages (Fig 2A). The bifurcation into separate cardiac fibroblast and cardiomyocyte lineages can clearly be observed in the PAGA plot (S5 Fig), which was created with the previously described cell type annotations, the re-computed neighborhood graph, and an edge weight threshold of 0.15. This PAGA embedding was used to define the two lineages used for downstream lineage isolation tasks, where all iPSC, mesoderm, cardiac mesoderm, and cardiac progenitor cells are assigned jointly to both lineages, while cardiomyocyte and cardiac fibroblast (terminal cell types) are unique to their corresponding lineage. Finally, four diffusion components were computed from the new neighborhood graph, and diffusion pseudotime was used to assign pseudotime values to cells from both cardiac lineages.

### Influence of batch on cell type composition

If a group of cells from the same batch are more likely to display similar cell type composition than cells from different batches, this could indicate that batch effects are confounding cell type annotation. However, since the experimental design places cells from the same cell line and differentiation stage in the same batch, it is important to disentangle similarity in cells due to the same cell line/ differentiation stage (a likely biological effect) from similarity in cells due to batch (a primarily technical effect; see S1 Fig). To do this, we grouped cells by cell line and differentiation day (called a 'sample' here), and summarized each group by their cell type

proportions. As indicated in **S1 Table**, each batch contains three different samples representing a different cell line/differentiation day combination. We focused on the impact of batch on terminal cell type composition by analyzing samples from collection day 3 that had been differentiating for 11 or 15 days. At these time points, terminal cell types (cardiomyocyte, CM, and cardiac fibroblast, CF) make up a majority of the cells present. Based on the experimental design, we have 19 pairs of samples such that one sample is at differentiation day 11, the other is at day 15, and the two samples belong to the same batch. If batch effects are causing cells from the same batch to receive similar terminal cell type labels, we would expect these sample pairs to display greater similarity of cell type proportion compared to a background of sample pairs from distinct batches matched for differentiation day. We computed pairwise differences in cell type proportion for both cardiomyocyte and cardiac fibroblast cell types, and compared these to a background (day 11 and day 15 samples from distinct batches) using a t-test. This analysis indicated that samples from the same batch are not more likely to display similar cell type composition than cells from different batches when controlling for cell line and differentiation day (CM $p = 0.286$, CF $p = 0.446$).

## Pseudobulk expression aggregation and normalization

Although the noisiness of single cell expression profiles necessitates aggregation across cells before dynamic eQTL calling, an improved understanding of the differentiation landscape allows us to pursue an aggregation strategy that mitigates the confounding impact of cellular composition differences and offers greater power than dynamic eQTL calling on bulk samples. Three pseudobulk aggregation schemes were used in this study:

1. *Chronological differentiation day binning*—This strategy is most directly comparable to bulk RNA-sequencing. Aggregation is performed by taking the sum of SCTransform-corrected counts from all cells from the same differentiation day and individual.

2. *Lineage subsetting*—Differentiation day binning was performed within each lineage separately. As evidenced by the PAGA graph, all cells up to the progenitor cell type (PROG) are assigned to both lineages, only cells from the terminal cell type (cardiomyocyte or cardiac fibroblast) are unique to one lineage or another.

3. *Lineage subsetting & pseudotime binning*—After lineage subsetting, cells are partitioned into 16 quantile bins according to pseudotime. We chose 16 bins in order to directly compare to our previous 16 time-point bulk experiment (see **S19 Fig**). Aggregation then consists of the sum of SCTransform-corrected counts from cells within the same cell line and pseudotime bin.

After pseudobulk aggregation, low-depth samples with library size less than 10,000 were filtered out. In this dataset, this cutoff was equivalent to removing samples with less than or equal to 50 cells. This removed 15 samples from the pseudotime-binned cardiac fibroblast lineage, all corresponding to pseudotime bins 14 or 15 (7 individuals were filtered out of both bins altogether, and one additional individual had only bin 15 filtered out). In the pseudotime-binned cardiomyocyte lineage, this filtered out 5 samples. Remaining samples underwent TMM normalization with singleton pairing through the *edgeR* package so that expression could be compared across samples for dynamic eQTL calling [69,70]. We then transform the TMM-normalized counts into compute counts per million (CPM) for each sample, and apply log normalization (with the *edgeR* package, which uses an approach to pseudocount addition that is adapted for library size). These logCPM expression values are used for QTL calling.

## Bulk expression normalization

In order to properly compare bulk RNA-seq data to our pseudobulk data, we reprocessed the bulk data from a previous experiment in a way that is intended to most closely match the logcpm pseudobulk expression. For this reason, we used transcripts per million (TPM) instead of previously used reads per kilobase of transcript, per million mapped reads (RPKM). For each sample, we first divided each gene's counts by the length in kilobase to compute reads per kilobase (RPK), and then fed these adjusted expression values into the same normalization pipeline as was used for pseudobulk counts (which are not biased by gene length)—TMM normalization with singleton pairing and logCPM adjustment, with the *edgeR* package. Since the input was reads per kilobase rather than counts, this gives logTPM expression values for use in QTL calling.

## Sample PCA

To identify primary sources of variation between samples, we ran principal component analysis (PCA) on the gene expression matrix for pseudobulk data. The first principal component is correlated with differentiation time (**S20 Fig**). For the top 10 PCs, we calculated the percent variance explained of each principal component by each technical factor recorded during sample collection (S14 Fig).

## Cell line collapsed PCA

To perform dynamic eQTL calling, we search for changes in gene expression over time that are correlated with a specific genotype. This can be confounded by broad differences between cell lines across the differentiation time course, such as differences in differentiation speed, lineage preference, or technical factors. For example, assume cell lines with genotype G at locus $i$ generally have increasing proportions of cardiomyocytes over time, while cell lines with genotype C at locus $i$ have increased proportions of cardiac fibroblasts over time. In this case, any gene whose expression is upregulated in cardiomyocytes will appear to have a dynamic eQTL at locus $i$, regardless of any cis-regulatory dynamics related to that gene, which constitute the intended focus of this study.

With single-cell data, we are able to more directly account for some of these factors, namely differentiation speed (with pseudotime binning) and lineage preference (with lineage subsetting). However, it remains useful to control for any broad cell line differences in this more unsupervised fashion, as any broad effects could drive false positive QTL detection.

We used a "cell line collapsed PCA" approach to identify such patterns across the entire time course [23]. To identify cell line collapsed PCs, we rearranged the gene expression matrix from the standard pseudobulk expression quantification such that each row represented expression from one cell line and each column represented a gene at a single time point. After standardizing each column to have zero mean and unit variance, we applied PCA to this matrix to learn a low dimensional representation. Each cell line has a shared loading across all time points, and PCs reflect broad differences in the way cell lines proceed through differentiation. For example, in bulk, it appears that the first cell line PC picks up on differences in differentiation speed between cell lines, while the second cell line PC picks up on differences in terminal cell type preference as defined as the highest total cell type proportion among days 10 to 15 (S9 Fig).

## Genotype data

We used previously collected and imputed genotype data for the 19 Yoruba individuals from the HapMap and 1000 Genomes Project [71]. For eQTL analyses, we filtered to variants with no missingness and a minor allele frequency of at least 0.1 across the 19 individuals present.

## Dynamic cis-eQTL test selection

We selected which genes to check for dynamic eQTLs based on the following filters:

- Gene must have at least 0.1 CPM in at least 10 bulk/pseudobulk samples

- Gene must have at least 6 counts (reads) in at least 10 samples

Both of these filters were applied separately for each aggregation scheme. We tested all variants within 50kb of the transcription start site of each gene. Transcription start sites were obtained from Gencode's release 37 [72], basic gene annotation, and matched to mapped genes by Ensembl gene ID. The total number of tests is presented alongside the number of dynamic eQTLs detected in Tables 1 and S2.

## Linear dynamic eQTLs using single-cell pseudobulk data

Linear dynamic eQTLs are cis-eQTLs whose effects are linearly modulated by differentiation time. We detected linear dynamic eQTLs with a Gaussian linear model that quantified the interaction between genotype and differentiation time on gene expression, while controlling for the linear effects of both genotype and differentiation time. We also controlled for linear effects of the first five cell line collapsed PCs (see below).

Following the method used in [23], we built a separate linear model for each tested variant-gene pair. Specifically, let $t$ denote the time point (or, for pseudotime binning, the median pseudotime value across cells constituting the pseudobulk sample) of the current sample, $c$ denote the cell line of the current sample, T denote the total number of time points, and C denote the total number of samples. $E \in R^{CxT}$ denotes the standardized expression matrix for the current gene, $G \in R^C$ denotes the dosage based genotype vector for the current variant, and $PC^K \in R^C$ denotes the Kth cell line collapsed PC vector. We modeled the expression levels as follows:

$$E_{ct} \sim N(\mu + \beta_1 G_c + \beta_2 t + \beta_3 PC_c^1 + \cdots + \beta_7 PC_c^5 + \beta_8 PC_c^1 t + \cdots + \beta_{12} PC_c^5 t + \beta_{13} G_c t, \sigma)$$

We used lmFit from the limma package to fit this model, and used a t-test to measure the significance of the genotype and time coefficient ($\beta_{13}$).

Bonferroni correction was applied to account for multiple SNPs being tested per gene, and Storey's q-value was used to control false discovery rates at the gene level, after selecting the most significant dynamic eQTL per gene [73].

Model diagnostics with partial regression and partial residual plots for the example linear dynamic eQTL in Fig 3A are shown in S12 Fig [74].

## Nonlinear dynamic eQTLs using single-cell pseudobulk data

To detect dynamic eQTLs whose effect size changes non-linearly with time, we used a second order polynomial basis function over time, which alters the above linear dynamic eQTL model as follows:

$$E_{ct} \sim N(\mu + \beta_1 G_c + \beta_2 t + \beta_3 t^2 + \beta_4 PC_c^1 + \cdots + \beta_8 PC_c^5 + \beta_9 PC_c^1 t + \beta_{10} PC_c^1 t^2 \ldots + \beta_{17} PC_c^5 t + \beta_{18} PC_c^5 t^2 + \beta_{19} G_c t + \beta_{20} G_c t^2, \sigma)$$

Once again, time is either time of collection, or median pseudotime of the sample. As before, we used lmFit from the limma package to fit this model, and this time used a similar t-test to measure the significance of the genotype and quadratic time coefficient ($\beta_{20}$). Multiple testing correction was applied as with linear dynamic eQTL calling.

Model diagnostics with partial regression and partial residual plots for the example nonlinear dynamic eQTL in Fig 3C are shown in S12 Fig [74].

## Control experiments

We assessed calibration of our dynamic eQTL calling methods with permutations. If we permute the time variable in the interaction term, we do not expect this term to properly capture interactions between genotype and time. For each variant-gene pair, we performed an independent permutation of the time variable in the interaction term, across all (cell line, day) samples. The results of this analysis are shown in S7 Fig.

As another check for confounding factors, we explored the possibility that broad differences between cell lines, such as variation in differentiation speed or trajectory preference, are driving false positive discoveries. If a pair of cell lines share properties such as trajectory preference that confound eQTL analysis, variants where those cell lines share genotype would be more likely to appear as dynamic eQTLs. If this is the case, we would expect pairwise correlation between cell lines according to genotype, across the top 200 dynamic eQTLs, to be higher than expected by chance (compared to a background set of random loci matched for minor allele frequency and distance to transcription start site), as these false dynamic eQTLs are not in fact picking up on distinct *cis* regulatory patterns but broad cell line patterns. This type of elevated correlation was not observed, suggesting that cell line PCA adequately controls for broad cell line differences (S8 Fig).

We also permuted pseudotime across all cells to investigate sensitivity to uncertainty surrounding pseudotime binning and variation in pseudobulk library sizes. For each of 100 cell-level permutations, we aggregated pseudobulk by the permuted pseudotime quantile bins, and performed dynamic eQTL calling. We used the absolute value of the resulting t-statistics to generate an empirical null distribution for each gene. The empirical p-values generally agreed with the original results up to the significance threshold imposed by the number of permutations performed (p = 0.01, -log10(p) = 2). These results do not suggest inflation of the nominal p-values from the original analysis, and were in fact less conservative than the p-values obtained using Student's t-distribution (S11 Fig).

## Simulations to examine type I errors due to 'double dipping'

We conducted simulations to evaluate potential type I error inflation caused by selective inference. We simulated gene expression data from the following linear mixed model:

$$Y_{ijk} = \beta_k G_{ik} + \alpha_k M_{ij} + a_{ik} + \epsilon_{ijk},$$

Here $Y_{ijk}$ is the expression of gene $k$ in cell $j$ of individual $i$, where $k = 1,\ldots,1000$, $j = 1,\ldots,100$ and $i = 1,\ldots,n$. The sample size $n$ is 10 or 20. We assumed one cis-eQTL per gene. To simulate the genotype $G_{ik}$, we first generated the minor allele frequency ($MAF_k$) from *Uniform*(0.1,0.5) and then generated $G_{ik} \sim binomial(2, MAF_k)$.

The other variables included genetic effect size $\beta_k$, cell maturity $M_{ij}$ and its effect size $\alpha_k$, individual-specific random effect $a_{ik}$ and error term $\epsilon_{ijk}$. They were generated from the following distributions:

$$\beta_k \sim N(0, \sigma_\beta^2), M_{ij} \sim N(0, 1), \alpha_k \sim N(0, \sigma_\alpha^2)$$

$$(a_{i1}, \ldots, a_{i,1000}) \sim N(0, \sigma^2 \Sigma), (\epsilon_{ij1}, \ldots, \epsilon_{ij,1000}) \sim N(0, \sigma_e^2 \Sigma),$$

Note that $(a_{i1},\ldots,a_{i,1000})$ are i.i.d. across individuals and $(\epsilon_{ij1},\ldots,\epsilon_{ij,1000})$ are i.i.d. across individuals and cells, but they are both correlated across genes. To construct a realistic correlation structure, we chose $\Sigma$ to be the correlation matrix of the expression of 1000 randomly selected genes from our pseudo bulk data. We fixed $\sigma_\alpha^2 + \sigma^2 = 0.3$ so that cell maturity and individual specific random effect explained 30% variance of expression and varied $\frac{\sigma_\alpha^2}{\sigma^2} = 0, 0.1, 0.5, 1, 2, 10$. We then generated the genetic effect size $\beta_k \sim N(0,0.1^2)$ or $N(0,0.4^2)$, corresponding to on average 0.4% or 6.3% variance of gene expression explained by genetic effects. The variance of the error term $\sigma_e^2$ was chosen so that the expression of each gene has unit variance.

We defined the pseudo time in this simulation study to be the first gene expression principal component (PC). We divided the cells into three equal pseudo time bins and averaged expression of the cells for each individual in each pseudo time bin into pseudo bulk expression ($\tilde{Y}_{ilk}$). We also calculated the average pseudo time for cells within each pseudo bulk sample, denoted by $t_{il}$. We tested two models for dynamic eQTL calling (fitted for each gene $k$ separately): 1) linear mixed model with individual-specific random effects $\tilde{Y}_{ilk} \sim G_{ik} + t_{il} + G_{ik}t_{il} + (1|\text{individual})$; 2) linear model $\tilde{Y}_{ilk} \sim G_{ik} + t_{il} + G_{ik}t_{il}$ without random effects. Type I error was calculated across 1000 genes (S10 Fig). The simulation suggests that a fixed-effect linear model for dynamic eQTL calling, as used in this study, was conservative in the presence of multiple measurements per individual and did not lead to type I error inflation. The more powerful linear mixed model did lead to moderate inflation.

## Correlation between bulk and pseudobulk data

We calculated the Pearson correlation of the normalized gene expression matrix from matched bulk RNA-seq data [23] with the normalized gene expression matrix from pseudobulk RNA-seq data. We observed a high correlation of gene expression values between bulk and pseudobulk samples of any given differentiation day (S21 Fig), and a consistent pattern of correlation for all cell lines (S22 Fig).

## Bulk dataset deconvolution using single cell data

Cell type deconvolution was performed using CIBERSORTx [53]. The method was first assessed for accuracy using pseudobulk data, where a ground truth is available. Cells from each annotated cell type were split into training (60% of cells) and testing (40%) groups. The annotated Seurat object was subset to training data, and the *FindAllMarkers* command was used to identify a subset of 404 genes for use in deconvolution. We removed genes that were not measured in bulk, leaving 317 genes for use in deconvolution. A gene expression signature matrix was created from exclusively the training data by taking the sum of SCTransform-corrected counts within each cell type. Normalization of the signature matrix was performed using edgeR: normalization factors were first computed with 'TMMwsp' method, then TMM-normalized counts were converted to counts per million. To assess the accuracy of this approach, we then used the same normalization pipeline to aggregate pseudobulk by sample for the testing data, where samples corresponded to a (cell line, differentiation day) combination (S13 Fig). To perform deconvolution of the bulk RNA sequencing data, we used the signature matrix described above and subset the bulk data to the 317 genes contained in the signature matrix.

## Cell type interaction eQTLs

To account for variable cell type composition in bulk RNA-seq data, rather than looking for cis-eQTLs whose effects are modulated by time (linear dynamic eQTLs), we looked for those whose effects are modulated by cell type proportion [22]. This mitigates the confounding

impact of lineage preference on dynamic eQTL calling, as well as differences in differentiation speed (to the extent that this is captured by cell type proportion). To do so, we replaced the time variable in the dynamic eQTL model with cell type proportion as follows:

$$E_{ct} \sim N(\mu + \beta_1 G_c + \beta_2 K_{ct} + \beta_3 PC_c^1 + \cdots + \beta_7 PC_c^5 + \beta_8 PC_c^1 K_{ct} + \cdots + \beta_{12} PC_c^5 K_{ct} + \beta_{13} G_c K_{ct}, \sigma)$$

Where $K_{ct}$ is the CIBERSORTx inferred cell type proportions in the sample. Separate models were built for each variant-gene pair, in each cell type except the 'unknown' cell type. We additionally explored a model in which we regressed out all cell type proportions (except the unknown cell type, as cell type proportions are constrained to sum to 1).

$$E_{ct} \sim N(\mu + \beta_1 G_c + \beta_2 K_{IPSC} + \ldots + \beta_7 K_{CM} + \beta_8 PC_c^1 + \cdots + \beta_{12} PC_c^5 + \beta_{13} PC_c^1 K_{ct} + \cdots + \beta_{17} PC_c^5 K_{ct} + \beta_{18} G_c K_{ct}, \sigma)$$

Note that while all fixed cell type proportion terms are included as covariates, there is only one interaction term for a single cell type proportion. Therefore, once again, separate models were fit for each variant-gene pair, in each cell type except 'unknown'. We found that regressing out additional cell types, not just the one included in the interaction term, led to detection of a greater number of genes with a cell type interaction eQTL (**S23 Fig**). To check whether these additional covariates were in fact introducing false positive associations between individuals, we measured the pairwise genetic correlation between cell lines among the top hits detected after regressing out additional cell type proportions. We then compared this to the genetic correlation among a set of hits detected before regressing out additional cell type proportions, matched for minor allele frequency. We did not see an increase in genetic correlation among significant tests introduced by incorporation of additional covariates (**S24 Fig**). However, we did observe a lower replication rate of this expanded set of interaction eQTLs among linear dynamic eQTLs ($\pi_1 = 0.69$ and $0.32$, respectively, compared to $0.84$ and $0.43$ under the first model).

We also explored including sample-level principal components as covariates in the linear model:

$$E_{ct} \sim N(\mu + \beta_1 G_c + \beta_2 U^1 + \ldots + \beta_7 U^5 + \beta_8 PC_c^1 + \cdots + \beta_{12} PC_c^5 + \beta_{13} PC_c^1 K_{ct} + \cdots + \beta_{17} PC_c^5 K_{ct} + \beta_{18} G_c K_{ct}, \sigma)$$

Where $U^1$ represents the first sample principal component, as opposed to $PC_c^1$, the first cell line principal component. Here, we again found that additional covariates led to an increased number of cell type interaction eQTLs detected (S23 Fig): for several cell types (pluripotent cells, mesoderm and progenitor) this figure continued to increase with up to 30 principal components regressed out. With the terminal cell types where more interaction eGenes were detected, the maximum number of hits detected occurred after regressing out 10 principal components. The replication rate among dynamic eQTLs decreased as the number of hits detected increased ($\pi_1 = 0.63$ and $0.30$ for cardiomyocyte and cardiac fibroblast, respectively, after 5 PCs were regressed out; $0.59$ and $0.38$ after 10; $0.64$ and $0.42$ after 20; $0.68$ and $0.44$ after 30). The results from fitting the first model are reflected in the main text.

## Static eQTL calling

The interaction testing frameworks described above are designed to test for significant associations between genotype and some environmental variable (pseudotime, real differentiation time, or cell type proportion). To specifically test for genetic variants associated with gene expression levels across all observed environmental contexts, it is not sufficient to perform a

similar hypothesis testing procedure on $\beta_1$, the coefficient associated with genotype. This coefficient measures the effect of genotype on expression specifically when the environmental variable is equal to zero, not across all environmental contexts [75]. As a result, estimates of this effect can be highly sensitive to the decision of how to center the environmental variable.

A more reliable way to identify eQTLs that are active across contexts is to perform eQTL calling in each context independently, and then pool information to infer patterns of sharing of regulatory effects across contexts, as done by the *mashr* package [46]. We performed regular eQTL calling in each cell type, and separately in each day, using the following model:

$$E_{ct} \sim N(\mu + \beta_1 G_c + \beta_2 PC_1 + \ldots + \beta_6 PC_5, \sigma^2)$$

Where $PC_i$ represents the $i^{th}$ sample principal component, computed from the single-context (day or cell type) gene expression data. We used both canonical and data-driven covariances, as described in [76], to fit the mash model to a random subset of 250,000 tests. To identify tests that are significant across all contexts, we first subset to the most significant test for each gene in any context. We then used the fitted mash model to count the number of conditions each of these effects is significant in, at a local false sign rate threshold of 0.05. This identified 183 static eQTLs using pseudobulk data aggregated by differentiation day, versus 147 static eQTLs using pseudobulk data aggregated by cell type.

## Overlap with published GTEx eQTLs

We used the GTEx v8 release to evaluate replication and overlap of our dynamic eQTLs with variants previously detected in adult tissues. To assess replication in each tissue, we used the qvalue package in R [52] to compute $\pi_1$ replication rates among all variant-gene pairs that were declared dynamic eQTLs that were also tested in GTEx. To determine the percentage of variant-gene pairs that were declared both dynamic eQTLs and significant *cis* eQTLs in GTEx, we incorporated *cis* eQTLs from all tissues.

## Supporting information

**S1 Table. Experimental Design.**
(XLSX)

**S2 Table. Comparison of nonlinear dynamic eQTL calling methods.** We report the number of nonlinear dynamic eGenes (genes with a significant nonlinear dynamic eQTL at gene-level q-value < = 0.05), for each of three aggregation schemes assessed. Total number of genes tested and total number of tests run are also reported.
(XLSX)

**S1 Fig. UMAP embedding colored by batch and sample.** (A) Cells are colored by batch (the experiment, collection day, and collection in which they were collected for sequencing). (B-E) Coloring by batch (B, D) and sample (C, E) shows that apparent batch effects are driven by similarity between cells of the same sample (cell line and differentiation day) within the batch, rather than the overall batch itself.
(TIF)

**S2 Fig. Principal component analysis of single cell data.** (Top Left) Principal components biplot for single cell data, colored by differentiation day. (Top Right) Principal components biplot for single cell data, colored by cell type. (Bottom) Violin plot of PC1 loadings on each cell, grouped by differentiation day.
(TIF)

**S3 Fig. Cell lines display differences in trajectory preference.** The force atlas embedding which was learned from all cells jointly is shown for each individual cell line, colored by cell type.
(TIF)

**S4 Fig. Cell cluster 6 appears to be an outlier cluster.** This group of cells which underexpresses cardiac markers from all stages of differentiation and overexpresses endoderm markers such as *APOA1* and *AFP* is picked up by the third principal component (top), and largely drives the variation behind the second diffusion component (bottom). The variation driven by relatively small population of cells interferes with reconstruction of biologically feasible trajectories, and was removed from downstream analysis.
(TIF)

**S5 Fig. PAGA identifies a bifurcation in cellular differentiation.** PAGA identifies a bifurcation into cardiomyocyte and cardiac fibroblast cell types after the cardiac progenitor stage.
(TIF)

**S6 Fig. Comparison of dynamic eQTL calling in bulk and pseudobulk.** (A) Number of dynamic eGenes that were detected in common between pseudotime-binned cardiomyocyte lineage, pseudotime-binned cardiac fibroblast, and previously collected bulk data. The majority of cardiomyocyte lineage dynamic eGenes overlap with at least one of the other two analyses. (B, C) Replication analysis of pseudobulk dynamic eQTLs in bulk [73]. (B) Distribution of nominal p-values from bulk data for the subset of gene-variant pairs that were identified as a dynamic eQTL in the pseudotime-binned cardiomyocyte lineage ($\pi_1 = 0.40$). (C) Distribution of nominal p-values from bulk data for the subset of gene-variant pairs that were identified as a dynamic eQTL in the pseudotime-binned cardiac fibroblast lineage ($\pi_1 = 0.13$).
(TIF)

**S7 Fig. Permutation analyses.** Permutation analyses (see *Control Experiments* in Materials and Methods) do not suggest substantial inflation in bulk (a), pseudotime-binned cardiomyocyte-subset pseudobulk (b), or pseudotime-binned cardiac fibroblast-subset pseudobulk (c). The p-values from this study are shown in blue, while those obtained from a permutation test are shown in red.
(TIF)

**S8 Fig. Genetic correlation across dynamic eQTLs.** In order to check whether broad cell line differences are driving false positive dynamic eQTLs, we compared genetic correlation among the top 200 linear dynamic eQTLs for bulk (top, left), and both pseudobulk lineages, cardiomyocyte (middle, left) and cardiac fibroblast (bottom, left), to genetic correlation among a set of background variants within 50kb of a gene, and matched for minor allele frequency (right).
(TIF)

**S9 Fig. Cell line PCA picks up on differences in the way cell lines progress through differentiation.** (Left) Inferred cell type proportions in bulk for each of the 19 cell lines, sorted by cell line PC1 loading. Focusing particularly on the proportions of iPSC and mesoderm cells, (blue and teal, respectively), it appears that cell line PC1 is picking up on differentiation speed, with cell lines with a higher PC1 score (lower subplots) differentiating slower than cells with a lower PC1 score. (Right) The second cell line PC score appears to separate cell lines based on their terminal cell type preference, cardiomyocyte or cardiac fibroblast, as defined by the most common cell type among differentiation days 10 to 15.
(TIF)

**S10 Fig. Selective inference simulations.** Simulations were performed to examine the impact of selective inference on type I error rates (*Simulations to examine type I errors due to 'double dipping'*). Under the generative model used, inflated type I error rates (bars exceeding the dashed line) were not observed when testing is performed using a linear model (blue).
(TIF)

**S11 Fig. Cell permutation analyses.** We generated an empirical null distribution by performing dynamic eQTL calling after permuting cell pseudotime upstream of pseudobulk aggregation. We compared the resulting empirical p-values (y-axis) to the nominal p-values from the original analysis (x-axis) in both lineages. We did not find evidence of inflation in the nominal p-values from the original analysis (instead, the contrast between the distributions suggests the nominal p-values may be overly conservative).
(TIF)

**S12 Fig. Regression Analysis.** (Top left) Linear dynamic eQTL, partial regression plot. For the linear dynamic eQTL example shown in Fig 3A
($\hat{\beta}_{g*t} = -2.13, p = 1.64 * 10^{-7}, q = 9.7 * 10^{-4}$), we obtained the residuals from regressing expression on all independent variables except the genotype $^*$ pseudotime interaction term (y-axis), and plotted these against the residuals from regressing the interaction term itself on all other independent variables (x-axis). The slope of the line shown measures the effect of the interaction between genotype and pseudotime after controlling for all other independent variables. (Top right) Linear dynamic eQTL, partial residuals plot. On the y-axis, $X_{g*t}\beta_{g*t}+res$, where $X_{g*t}$ is genotype*time for a cell line/pseudotime bin pseudobulk sample, $\beta_{g*t}$ is the estimated coefficient for the genotype*time interaction term, and *res* are the residuals from the fitted linear dynamic eQTL model. On the x-axis is $X_{g*t}$. (Bottom) Similar partial regression and partial residuals plots (respectively) for the nonlinear dynamic eQTL shown in Fig 3C
($\hat{\beta}_{g*t^2} = 30.1, p = 1.26 * 10^{-8}, q = 1.3 * 10^{-3}$), where the interaction term of interest is between genotype and pseudotime squared.
(TIF)

**S13 Fig. CIBERSORTx assessment in pseudobulk.** Assessment of CIBERSORTx performance in pseudobulk, where 'ground truth' is available. CIBERSORTx-estimated cell type proportions from differentiation day-binned pseudobulk data for three cell lines is shown at left ('inferred'), compared to true cell type proportions ('true', right), as determined by the cell type annotation approach described in the supplement.
(TIF)

**S14 Fig. PCs percent variance explained by technical factors in single cell data.** (a) Variance explained of each gene expression principal component (1–10) for pseudobulk samples aggregated by cell line and differentiation day using recorded covariates, including: percent cells beating (visually assessed), differentiation day, collection day, culture confluence, cell morphology (visually assessed), and cellular debris. (b) Variance explained of principal components for pseudobulk samples aggregated by cell line pseudotime bin for cardiac fibroblast (CF, left) and cardiomyocyte (CM, right) lineages. Technical covariates shown are cell line, library size, median pseudotime, number of cells, and the normalization factor used for TMM normalization, from the edgeR package (see Materials and Methods).
(TIF)

**S15 Fig. Number of UMIs, genes, and percent mitochondrial reads per cell in single cell data, by day.** Distribution of the number of Unique Molecular Identifiers (UMIs) per cell, number of genes per cell, and the percent mitochondrial reads per cell in full single cell dataset,

prior to (top row) and after (bottom row) filtering as described in Materials and Methods (*RNA-seq quantification*). X-axis separated by differentiation day.
(TIF)

**S16 Fig. Number of UMIs, genes, and percent mitochondrial reads per cell in single cell data, by individual.** Distribution of the number of Unique Molecular Identifiers (UMIs) per cell, number of genes per cell, and the percent mitochondrial reads per cell in full single cell dataset, prior to (top row) and after (bottom row) filtering as described in Materials and Methods (*RNA-seq quantification*). X-axis separated by cell line.
(TIF)

**S17 Fig. Number of cells per sample.** Number of cells per collected sample following filtering described in Materials and Methods (*RNA-seq quantification*).
(TIF)

**S18 Fig. UMAP embedding with outlier clusters removed.** As in Fig 1C, a UMAP embedding of the single cell dataset colored by cell type, except with outlier clusters removed.
(TIF)

**S19 Fig. Dynamic eQTL detection rates across multiple bin sizes.** Y-axis shows the number of significant linear dynamic eGenes (genes with a dynamic eQTL, q<0.05) for a variety of numbers of pseudotime quantile bins (x-axis) for both the cardiac fibroblast (pseudobulk-cf, left) and cardiomyocyte (pseudobulk-cm, right) lineages.
(TIF)

**S20 Fig. PCA on pseudobulk and bulk samples identifies differentiation progress as primary source of variation.** PCA on bulk (row 1), single cell data aggregated into pseudobulk by differentiation day/ individual (row 2), cardiomyocyte lineage-specific single cell data aggregated into pseudobulk by pseudotime / individual (row 3), and cardiac fibroblast lineage-specific single cell data aggregated into pseudobulk by pseudotime/individual (row 4). Samples colored on a gradient by (left column) differentiation day or pseudotime bin, or (right column) cell line.
(TIF)

**S21 Fig. Correlation of bulk and pseudobulk data by day.** Pearson correlation between single-cell pseudobulk data and bulk RNA-seq data [23] for each individual; panels separated by differentiation day.
(TIF)

**S22 Fig. Correlation of bulk and pseudobulk data by individual.** Pearson correlation between single-cell pseudobulk data and bulk RNA-seq data [23] for each differentiation day; panels separated by individual.
(TIF)

**S23 Fig. Impact of additional covariates on interaction eQTL calling.** We examined the impact of regressing out additional covariates from the interaction eQTL model, and found an increase in the number of genes with a dynamic eQTL, as well as a decrease in the replication rates in bulk dynamic eQTLs (Materials and Methods) for both regression of cell type proportions (top) and up to 30 principal components (bottom).
(TIF)

**S24 Fig. Genetic correlation across cell type interaction eQTLs.** We compared genetic correlation among 200 cardiac fibroblast cell type interaction eQTLs detected exclusively after

regressing out additional cell type proportion covariates (a), compared to 200 interaction eQTLs, detected before controlling for cell type proportions (b). We similarly computed genetic correlation among 200 cell type interaction eQTLs discovered only after regression of 5 (c), 10 (d), 20 (e), and 30 (f) sample principal components.
(TIF)

## Acknowledgments

We thank Natalia Gonzales for providing feedback on the manuscript, and the lab of Anindita Basu for their support with Drop-seq.

## Author Contributions

**Conceptualization:** Yoav Gilad, Alexis Battle.

**Data curation:** Reem Elorbany.

**Formal analysis:** Reem Elorbany, Joshua M. Popp, Guanghao Qi.

**Funding acquisition:** Yoav Gilad, Alexis Battle.

**Investigation:** Reem Elorbany, Katherine Rhodes, Kenneth Barr.

**Methodology:** Reem Elorbany, Joshua M. Popp, Benjamin J. Strober.

**Supervision:** Yoav Gilad, Alexis Battle.

**Visualization:** Joshua M. Popp.

**Writing – original draft:** Reem Elorbany, Joshua M. Popp.

**Writing – review & editing:** Reem Elorbany, Joshua M. Popp, Katherine Rhodes, Benjamin J. Strober, Kenneth Barr, Guanghao Qi, Yoav Gilad, Alexis Battle.

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
