## [Decision Letter · Decision Letter 0]

3 Aug 2021

Dear Dr Battle,

Thank you very much for submitting your Research Article entitled 'Single-cell sequencing reveals lineage-specific dynamic genetic regulation of gene expression during human cardiomyocyte differentiation' to PLOS Genetics.

The manuscript was fully evaluated at the editorial level and by independent peer reviewers. The reviewers appreciated the attention to an important problem, but raised some substantial concerns about the current manuscript. Based on the reviews, we will not be able to accept this version of the manuscript, but we would be willing to review a much-revised version. We cannot, of course, promise publication at that time.

If you decide to revise the manuscript for further consideration at PLOS Genetics, please aim to resubmit within the next 60 days, unless it will take extra time to address the concerns of the reviewers, in which case we would appreciate an expected resubmission date by email to plosgenetics@plos.org.

[LINK]

We are sorry that we cannot be more positive about your manuscript at this stage. Please do not hesitate to contact us if you have any concerns or questions.

Yours sincerely,

Mingyao Li

Associate Editor

PLOS Genetics

Hua Tang

Section Editor: Natural Variation

PLOS Genetics

Reviewer's Responses to Questions

**Comments to the Authors:**

Reviewer #1: Elorbany and coauthors carried out scRNA-seq on 19 cell lines at 7 different time points and identified lineage-specific dynamic eQTLs during cardiomyocyte differentiation. The paper is easy to follow, but there are a few points that need to be addressed before this is suited to be published in PLOS Genetics.

Major:

1. The authors identified two major lineages, in concordance with their previous findings. The differentiation process involves transient cell states, instead of discrete cell types. I am concerned about the clustering analysis/results shown in Figure 1. A trajectory analysis will be more appropriate – granted, the authors inferred the pseudotime in Figure 2, but the results in the first figure fall in the domain of hard clustering and may raise questions. In addition to trajectory reconstruction, the authors can also consider “soft” clustering instead. For example, the PAGA method that was referenced [44].

2. There are two lineages (CM and CF) identified. How is pseudotime inferred and assigned for these two lineages given there is a split? Figure 2b does not clearly show the bifurcation pattern, and the pseudotime seems to be monotonically increasing along the reduced dimensions. Did the authors carry out two lineage-specific eQTL analyses using pseudotime x Genotype as a covariate?

3. While the authors show the number of significant gene-SNP pairs in Table 1, it would be more informative to show how they compare and contrast via, for example, a venn diagram. How many of the SNPs identified by the pseudobulk are also identified by the bulk analysis with more timepoints? How do they compare against the original bulk analysis? The authors mentioned, “This variant is not detected as a dynamic eQTL without lineage subsetting or pseudotime binning.” Is it detected by lineage subsetting (the original Science paper identified two clusters of cell lines)? In this case, the authors could more clearly emphasize the utility of the single-cell data.

4. The authors carried out close to 2 million tests, and I am concerned about the power issue. For the two cases in Figure 3a and Figure 3c, visualization of the regression analysis is needed to show the estimated beta_13 and beta_20 with their estimates and nominal p-values (e.g., using partial residuals).

Minor:

1. It would be good to include a reduced dimension plot with colors corresponding to batch. I understand that the authors carefully designed the experiment, but there are 131 samples – is batch effect corrected for?

2. It would be more appropriate to reduce the dimensions and mark the two lineages after removing the unknown cells, as was done in Figure 2.

3. The figure legends and axis labels are too small to be read.

4. In Figure 3A, boxes for GG at timepoint 14 and 15 are missing.

Reviewer #2: The authors perform single cell analysis of 19 donors across 7 time points in order to examine dynamic genetic regulation during differentiation of cardiomyocytes. They find both linear and non-linear dynamic eQTLs, a number of which would not have been found without the use of single cell resolution, which enabled pseudotime reconstruction and lineage subsetting. The disentangling of cell line specific differentiation speed is a key result, and the detailed analysis presented here provides a framework for future studies. The authors then compare their results to a previous bulk eQTL time series dataset and to GTEx. Finally, the authors use cell type proportions per cell line to deconvolve the bulk samples and detect cell type interaction eQTLs. The dataset and analysis presented here are both valuable for increasing the understanding of dynamic genetic regulation of these cell types, as well as for the discussion of technical issues and optimizations that should be taken into account when performing single-cell QTL studies from iPSC.

Major comments:

- "We collected single-cell data using a balanced study design in which each collection included three individuals at three unique differentiation time points." Please provide the study design as a Supp Table or additional data table. Do the collection groups from the study design correlate with the two clusters of cell lines with respect to the bifurcation in cell fate?

- A number of results are not presented in Results in the order I would have expected. The simulation to assess double dipping from latent variables in the linear model formula (S18) doesn't follow directly the section on the linear model but shows up in the Discussion. Likewise the permutation analysis (S10) is only mentioned in Methods. These seem to be key control experiments that should be at the least briefly mentioned in main text.

Minor comments:

In line 209, it is described how and why single cells were aggregated within pseudotime bins separately along each lineage. This facilitates analysis, allows for use of well-developed tools for bulk profiles, and as described in the text makes analysis perhaps more robust. Can the authors discuss any potential downsides to this approach compared to a model at the single cell level using the continuous inferred pseudotime variable?

From line 221, the dynamic eQTLs are presented, e.g. 357 eGenes in CM and 903 in CF. These are the significant results from interaction of genotype and differentiation time. How many stable eGenes were found? That is the significance of beta_1 in line 703. Or are stable, genetically regulated eGenes not possible to detect with their analysis framework?

"Ultimately, our lineage subsetting and pseudotime approach revealed more dynamic eQTLs than were previously identified...The increased detection rate may stem from...", this is technically true but the 3% increase might be better described as "nearly the same" or "slightly more". At first I missed that this was the increase being described.

In Fig 3a and 3c, it would be useful to know the number of samples per genotype x timepoint, if there is a condensed way to include this information on the plot. At the least, knowing the mean number of samples per genotype would be useful.

Line 266: recommend to cite qvalue here to give background to the pi_1 statistic.

Line 270 "by searching directly for dynamic effects across tissues rather than within a single tissue in isolation": Suggest rewording "across cell types" rather than "across tissues" as in principle the analysis presented does not require a multi-tissue system.

Line 298: any hypothesis as to why the CF replication was much lower than the CM replication as estimated by pi_1?

Line 301: how many of the ieQTLs were lineage specific?

Line 385: "infeasible": for the sake of specificity, do the authors mean that latent variable inference is computationally intensive and thus would be a bottleneck with current algorithms?

Line 389-391: the fixed-effect linear model may nevertheless lead to loss in sensitivity as it is quite a bit overly conservative (as mentioned in Methods).

Line 661: "Each cell line has a shared loading across all time points, and PCs reflect trajectories across all genes" I followed the cell line collapsed PCA up until this point. Rows represent cell lines and columns represent gene x time points. So the loadings for each PC are of dimension genes x time points (~38k x 7)? I wasn't clear how this reflects trajectories.

Line 732: "we checked whether dynamic eQTLs were enriched for genotypes shared between any particular pair of individuals (suggesting broad individual differences could be driving the dynamic eQTLs, Fig. S10)." Didn't understand what it meant for "eQTLs to be enriched for genotypes", can the authors be more specific? And Fig S10 is the permutation analysis only, was there another Fig to be referenced here (S15)?

Line 767 - extra line break.

Fig S10 "do not suggest inflation", would suggest "do not suggest substantial inflation" as there is some inflation of small p-values for (a) and (c) panels here.

Reviewer #3: Attached review as an attachment.

**Have all data underlying the figures and results presented in the manuscript been provided?**

Reviewer #1: Yes

Reviewer #2: Yes

Reviewer #3: Yes

PLOS authors have the option to publish the peer review history of their article (what does this mean?). If published, this will include your full peer review and any attached files.

Reviewer #1: No

Reviewer #2: **Yes: **Michael Love

Reviewer #3: No

---

## [Decision Letter · Decision Letter 1]

21 Dec 2021

Dear Dr Battle,

We are pleased to inform you that your manuscript entitled "Single-cell sequencing reveals lineage-specific dynamic genetic regulation of gene expression during human cardiomyocyte differentiation" has been editorially accepted for publication in PLOS Genetics. Congratulations!

Yours sincerely,

Mingyao Li

Associate Editor

PLOS Genetics

Hua Tang

Section Editor: Natural Variation

PLOS Genetics

Comments from the reviewers (if applicable):

Reviewer's Responses to Questions

**Comments to the Authors:**

Reviewer #1: The authors have addressed my previous concerns.

Reviewer #2: The authors have addressed all of my previous concerns, in particular assessing whether the design would contribute to cell fate differences (not significant when assessing pairs of batched samples relative to background pairs of samples). A number of additional analyses were performed to answer questions from my first review, and these questions have been directly answered in every case. Figure S9 demonstrates the intuition behind the cell line PCs representing differentiation speed and terminal cell type preference.

Note that the design is listed as Table S1 but was uploaded as Table S2.

Reviewer #3: I appreciate the authors' effort to address my questions and concerns. I felt the authors adequately addressed all points.

**Have all data underlying the figures and results presented in the manuscript been provided?**

Reviewer #1: None

Reviewer #2: Yes

Reviewer #3: None

PLOS authors have the option to publish the peer review history of their article (what does this mean?). If published, this will include your full peer review and any attached files.

Reviewer #1: No

Reviewer #2: **Yes: **Michael Love

Reviewer #3: No

**Data Deposition**

http://datadryad.org/submit?journalID=pgenetics&manu=PGENETICS-D-21-00889R1

**Press Queries**

---

## [Editor Report · Acceptance letter]

14 Jan 2022

PGENETICS-D-21-00889R1 

Single-cell sequencing reveals lineage-specific dynamic genetic regulation of gene expression during human cardiomyocyte differentiation 

Dear Dr Battle, 

We are pleased to inform you that your manuscript entitled "Single-cell sequencing reveals lineage-specific dynamic genetic regulation of gene expression during human cardiomyocyte differentiation" has been formally accepted for publication in PLOS Genetics! Your manuscript is now with our production department and you will be notified of the publication date in due course.

With kind regards,

Zsofia Freund

PLOS Genetics

On behalf of:
